# A Kinetic Model for Anaerobic Digestion and Biogas Production of Plant Biomass under High Salinity

**DOI:** 10.3390/ijerph19116943

**Published:** 2022-06-06

**Authors:** Jing Wang, Bing Liu, Meng Sun, Feiyong Chen, Mitsuharu Terashima, Hidenari Yasui

**Affiliations:** 1School of Municipal and Environmental Engineering, Shandong Jianzhu University, Jinan 250101, China; jwang@sdjzu.edu.cn; 2Resources and Environment Innovation Research Institute, Shandong Jianzhu University, Jinan 250101, China; ctokyo@hotmail.com; 3Faculty of Environmental Engineering, The University of Kitakyushu, 1-1, Hibikino, Wakamatsu, Kitakyushu 808-0135, Japan; m-sun@kitakyu-u.ac.jp (M.S.); m-terashima@kitakyu-u.ac.jp (M.T.); hidenari-yasui@kitakyu-u.ac.jp (H.Y.)

**Keywords:** ADM1, high salinity, kinetics, methane fermentation, sulfate reduction

## Abstract

The aim of this study is to evaluate the anaerobic digestion and biogas production of plant biomass under high salinity by adopting a theoretical and technical approach for saline plant-biomass treatment. Two completely mixed lab-scale mesophilic reactors were operated for 480 days. In one of them, NaCl was added and the sodium ion concentration was maintained at 35.8 g-Na^+^·L^−1^, and the organic loading rate was 0.58-COD·L^−1^·d^−1^–1.5 g-COD·L^−1^·d^−1^; the other added Na_2_SO_4_–NaHCO_3_ and kept the sodium ion concentration at 27.6 g-Na^+^·L^−1^ and the organic loading rate at 0.2 g-COD·L^−1^·d^−1^–0.8 g-COD·L^−^^1^·d^−1^. The conversion efficiencies of the two systems (COD to methane) were 66% and 54%, respectively. Based on the sulfate-reduction reaction and the existing anaerobic digestion model, a kinetic model comprising 12 types of soluble substrates and 16 types of anaerobic microorganisms was developed. The model was used to simulate the process performance of a continuous anaerobic bioreactor with a mixed liquor suspended solids (MLSS) concentration of 10 g·L^−1^–40 g·L^−1^. The results showed that the NaCl system could receive the influent up to a loading rate of 0.16 kg-COD/kg-MLSS·d^−1^ without significant degradation of the methane conversion at 66%, while the Na_2_SO_4_–NaHCO_3_ system could receive more than 2 kg-COD·kg^−1^-MLSS·d^−1^, where 54% of the fed chemical oxygen demand (COD) was converted into methane and another 12% was observed to be sulfide.

## 1. Introduction

Semi-arid regions in Central Asia suffer from reduced crop productivity due to salinization caused by climate change and poor irrigation control [1,2,3,4]. The main salts in the salinized soil are Na_2_SO_4_ and/or NaCl; the proportions of Na_2_SO_4_ and NaCl together, Na_2_SO_4_ individually, and NaCl individually are 62%, 28%, and 10%, respectively, in the salinized soil of Uzbekistan [5]. The guidelines of Food and Agriculture Organization of the United Nations (FAO) recommend phytoremediation using halophytes to improve soil conditions [6]. However, the treatment of the halophytes generated after harvesting involves two problems. First, the halophytes rot during storage [7] because the moisture content is high (ca. 75–90%) [8,9]. Second, the perennial halophytes Campanulaceae have a high lignocellulose content. Various types of halophytes naturally occur in Uzbekistan, and some species are capable of extracting salts from the salinized soil into the vacuoles in their cells [5,6,10,11]. Such plants may be used for the phytoremediation of the arid land in Central Asia [11].

Biomass is an important source of bioenergy. Hence, the anaerobic digestion of agricultural biomass and other organic waste to produce biogas has attracted considerable attention [12]. Lignocellulosic materials such as corn stover and wood grass have been shown to be biodegradable for methane fermentation [13,14,15,16], where a conversion efficiency of 10–80% has been observed depending on the species. However, the digestion pathway and model have not been studied. Ward et al. [17] highlighted the successful anaerobic digestion of halophytic microalgae under high-density salinity (70 g-NaCl·L^−1^); thus, a methane-production pathway was confirmed and the need for pretreatment was negated. Meanwhile, it is necessary to consider the effect of sulfate ions owing to their high density in semi-arid regions and halophyte bodies.

Batstone et al. developed a simple extension of the anaerobic digestion model No. 1 (ADM1) [18] for sulfate reduction [19]. However, sulfate-reducing bacteria (SRB) can use different substrates as electron donors [20,21,22,23] that may compete with *Methanogens* [24,25,26,27]. Therefore, it is necessary to develop an applicable model for anaerobic plant digestion under high salinization, including sulfate reduction.

This study evaluates the anaerobic fermentation of plant biomass in a lab-scale mesophilic anaerobic digestion (AD) reactor under saline conditions to examine the process performance of a continuous anaerobic system. Furthermore, experimentally obtained datasets were simulated using a developed biological model to estimate the microbial activity in the reactor. The ADM1 model was extended by oxalate digestion and sulfate reduction to reveal the anaerobic biomethanation effect of plant biomass under high salinity.

## 2. Materials and Methods

### 2.1. Dry Fodder Grass Biomass Components

In preliminary work (to be published), along with the chemical analysis of proteins and lipids, the dry fodder biomass chemical oxygen demand (COD) from the composites in the disintegration step was estimated to have the following composition: 61% carbohydrates, 10.8% proteins, 0.1% lipids, 1.1% oxalate, and 27% inert (lignin); lignin is non-biodegradable, whereas the other components are biodegradable (see Figure 1). All the estimated fractions were directly used as input for the model constructed in this study, except that 1% soluble inert fractions were subtracted from the carbohydrates, which were calibrated from the analysis of soluble lignin compounds in the digestate.

In the preliminary experiment, the electrical conductivity immediately increased after mixing the dried halophytes (three typical species of halophytes) with water, and the mixture became stable after 2 h in a stirred beaker (datasets not shown). This implies that the salts in the vacuole were released into the liquid, which justified the preparation of the “synthetic halophyte from grass and salts”.

### 2.2. Continuous Experiment

To investigate anaerobic plant digestion performance under high salinization, two 4 L (effective capacity) stirred bioreactors (MDL-1000, BEM, Tokyo, Japan) at agitation speed of 50 rpm (see Figure 2) were fed for a sludge retention time (SRT) of 40 days with dried annual fodder grass (158 g-COD·L^−1^ or 214.75 g-grass·L^−1^), which was well pulverized and sifted through a wire sieve with an aperture diameter of 224 μm. One reactor was fed biomass mixed with Na^+^ derived from NaCl, and the concentration was 70 mg-NaCl·g^−1^, i.e., it was two times the concentration of seawater, which has an average concentration of 35 mg-NaCl·g^−1^. The other reactor was fed biomass mixed with Na^+^ derived from Na_2_SO_4_ and NaHCO_3_. In the Na_2_SO_4_–NaHCO_3_ system, FeCl_2_ was added to adjust the pH and prevent the formation of toxic H_2_S by producing solid ferrous sulfide during the operation process. The concentrations of Na^+^ in the two reactors were maintained at 35.8 g-Na^+^·L^−1^ and 27.6 g-Na^+^·L^−1^, with osmotic pressures of 80.72 atm and 62.66 atm, respectively, at 35 °C, according to calculations using the OLI Analyzer software (OLI Systems, Inc., Parsippany, NJ, USA). Furthermore, trace elements (1 mg-Ni·L^−1^ and 1 mg-Co·L^−1^) were added as supplemental nutrients. The feed substrates and the conditions of the two systems are summarized in Table 1.

In the preliminary incubation stage, the two reactors were operated for 600 days at low volumetric organic-loading rates (OLRs) (NaCl system, 0.58 g-COD·L^−1^·d^−1^; Na_2_SO_4_–NaHCO_3_ system, 0.2 g-COD·L^−1^·d^−1^). Then, the OLRs of the two systems were changed in a stepwise manner. For the NaCl system, the OLR was in the range of 0.58 g-COD·L^−1^·d^−1^–1.50 g-COD·L^−^^1^·d^−1^; for the Na_2_SO_4_–NaHCO_3_ system, it was in the range of 0.20 g-COD·L^−1^·d^−1^–0.80 g-COD·L^−1^·d^−1^. As the organic addition volume varied with the OLR, a calculated sludge amount was taken every day; then, 100 mL samples from the two reactors were centrifuged (10,000 rpm, 5 min), and the supernatant and solid were used for analysis. The superfluous sludge was centrifuged, and the solid was returned to the reactors. Thus, the SRT and OLR could be ensured during operation. The methane productions of the two jar fermenters were measured using a gas counter (MGC-1, Ritter) after removing carbon dioxide using calcium oxide.

### 2.3. Analytical Procedures

#### 2.3.1. Anion Concentrations

The volatile fatty acids below C6, oxalate, and inorganic anions were detected using an ion chromatography system equipped with an IonPac AS11-HC column (ICS-1000, Thermo Fisher Scientific Inc., Waltham, MA, USA). Further, the elute flow rate was set to 1 mL·min^−1^ at 35 °C with 4 mol·L^−1^ of hepta-fluoro-butyric acid for the organic acids and 4 mol·L^−1^ of KOH for the inorganic anions.

#### 2.3.2. Soluble Organic Concentrations

The soluble carbohydrate (soluble total sugar) concentration was calorimetrically analyzed using the phenol–sulfuric acid method [28] with a glucose standard (Kishida chemical, Osaka, Japan).

The soluble protein concentration was calculated on the basis of the soluble K-N concentration excluding NH_x_-N with an egg albumin standard (Kishida Chemical, Osaka, Japan).

The soluble lipid concentrations were measured using the Soxhlet extraction method according to #5520 D in Standard Methods [29].

For the soluble lignin concentration, the Lowry–Folin method [30] could detect both protein and polyphenolic compounds (soluble lignin). The lignin concentration was calculated using the Lowry–Folin method by subtracting the protein concentration. The conversion factor of the soluble lignin concentration in the Lowry–Folin method was determined using an alkali-extracted lignin standard (Tokyo Chemical Industry, Tokyo, Japan).

#### 2.3.3. Continuous Operation (Regular Measurement and Calculation)

The methane production rate (MPR), volatile fatty acids (VFAs), volatile suspended solids (VSS), soluble total organic carbon (TOC) (TOC-VCSN, Shimadzu, Kyoto, Japan), and pH were measured regularly [29]. The particulate and soluble COD concentrations were calculated from the VSS concentration using 1.19 g-COD particulate·g^−1^-VSS (192 g-O_2_ to completely oxidize 162 g-carbohydrate, (C_6_H_10_O_5_)_n_) and 2.67 g-COD soluble·g^−1^-TOC soluble (32 g-O_2_·12 g-C^−1^).

### 2.4. Dynamic Simulation

A dynamic simulation of the continuous reactor response was conducted by focusing on the chronological change in the methane production rate, particulate COD, and soluble concentrations (soluble COD, dominant organic acids (acetate and propionate)). Based on the decomposition of the organics, an additional sulfate reducing reaction using an organic as an electron donor, and the accumulation of intermediates (acetate and propionate), the kinetics for the individual organics were specified using a biochemical process map (see Figure 3). For this purpose, anaerobic digestion model no.1 (ADM1) [18] and an extended sulfate reducing model were adopted in this study. As the material balance of the model was COD-based, the COD/DOC factors (g-COD·g-DOC^−1^·g^−1^) for each substance were prepared as 2.67, 3.00, and 2.92, respectively, to calculate each soluble composite material concentration (carbohydrate, protein, and lignin) for assuming the elemental compositions of carbohydrate: (CH_2_O)_n_, protein: (C_4_H_9_O_2_N)_n_, and lignin: (C_31_H_34_O_11_)_n_ [31].

A process simulator (GPS-X ver.6.4, Hydromantis Environmental Software Solutions, Inc., Hamilton, ON, Canada) was used to program the model and numerically solve the set of differential equations. The components of plant biomass can be decomposed in five steps (disintegration, hydrolysis, acidogenesis, acetogenesis, and methanogenesis) [18,19] under anaerobic high-salinity conditions by methane-producing microorganisms, which outperform sulfate-reducing microorganism in terms of the VFAs and hydrogen consumption (see Figure 3).

This model differs from the general ADM1 model in two aspects in terms of the model structure. One is the release, uptake, and degradation of oxalic acid in the vacuoles. The other is that the SRB and methanogens compete for electron donors [32].

A model including the methane-fermentation and sulfate-reduction processes was constructed. The process expression of sulfate reduction was the same as that of methane fermentation (see Table 2 and Table 3).

The values of the maximum specific uptake rates (k_m_) and the half-saturation coefficients (K_S_) of the individual degraders in the acidogenesis and acetogenesis steps were modified to meet the soluble COD accumulation. Owing to salt inhibition, the values of k_m_ and K_S_ are predicted to be less than the default values because the Na^+^ concentration was maintained at 27.6 g-Na^+^·L^−1^ or 35.8 g-Na^+^·L^−1^ during operation.

First-order type (Equation (1) in r_1_–r_4_ and Equation (2) in r_22_–r_37_) and Monod-type (Equation (3) in r_5_–r_11_ and r_15_–r_21_) rate equations were used to express the reaction rates of the model (see Table 2 and Table 3). The mathematical formulas are as follows.
(1)$$\rho_{j}=k_{\mathrm{process}}\times X_{i}$$
(2)$$\rho_{j}=b_{\mathrm{biomass}}\times X_{B}$$
(3)$$\rho_{j}=k_{m}\times \frac{S_{i}}{K_{s}+S_{i}}\times X_{B}$$
where ρ_j_ is kinetic rate of process j, kgCOD·m^−3^·d^−1^; k_process_ is first-order parameter for disintegration and hydrolysis, d^−1^; and X_i_ is concentration of particulate components i kgCOD·m^−3^; b_biomass_ is decay coefficient for biomass, d^−1^; X_B_ is concentration of biomass, kgCOD·m^−3^; k_m_ is Monod maximum specific uptake rate, kgCOD_S·kgCOD·X_B_^−1^·d^−1^; S_i_ is concentration of the soluble components i, kgCOD·m^−3^; and K_S_ is Monod half-saturation coefficient, kgCOD·m^−3^.

Considering the propionate accumulation in the operation process, as with other VFAs, propionate can inhibit methanogenic activity. To overcome this problem, a non-competitive propionate inhibition function ^n^ with a power factor (n) was created in r_12_, r_13_, r_14_ (see Table 2 and Table 3). The inhibition equation obtained is shown in Equation (4), in which a power factor n was included.
(4)$$\rho_{j}=\frac{K_{I}^{n}}{K_{I}^{n}+S_{\mathrm{Ii}}^{n}}\;$$
where ρ_j_ is kinetic rate of process j, kgCOD·m^−3^·d^−1^; K_I_ is inhibition constant, kgCOD·m^−3^; and S_Ii_ is inhibitory component i, kgCOD·m^−3^.

The kinetics of the biological reaction model were obtained by reproducing the experimental datasets.

## 3. Results and Discussion

### 3.1. Continuous Experiment

#### 3.1.1. Effect of High Salinity in the NaCl System

The methane production was conducted as shown in Figure 4 with the OLR ranging from 0.58 g-COD·L^−1^·d^−1^ to 1.50 g-COD·L^−1^·d^−1^ from the beginning to day 289 because the influent-digested COD was converted into methane gas without acid inhibition, as the effluent VFA concentration was maintained at a low level. Owing to the appearance of high propionate concentrations from day 250, the feeding of the influent was from day 290 to 320. The detection of VFAs (propionate and acetate) from day 250 to 320 could be attributed to the propionate and acetate non-methanization at high sodium concentrations. In previous studies [33,34], it was concluded that the threshold for the adaptation of the anaerobic sludge to the degradation of VFAs is lower for propionate than for acetate, at 21.5 g·L^−1^ of sodium concentration. In this study, 70 g·L^−1^ of sodium was used, as the propionate-utilizing and acetate-utilizing microorganisms were estimated to be more sensitive during operation, with the OLR increasing under the high salinity; the unconverted COD, especially acetate and propionate, was assumed to remain in the reactor. From day 320, the OLR recovered to 0.8 g-COD·L^−1^·d^−1^ and changed to 0.75 g-COD·L^−1^·d^−1^ from day 330 to day 480. The attained methane conversion efficiency based on the overall COD was 66% at an OLR of 0.75 g-COD·L^−1^·d^−1^.

#### 3.1.2. Competition of SBR in the Na_2_SO_4_–NaHCO_3_ System

In the Na_2_SO_4_–NaHCO_3_ system, as shown in Figure 5, methane gas was continuously produced at a controlled OLR between 0.2 g-COD·L^−1^·d^−1^ and 0.8 g-COD·L^−^^1^·d^−1^ for 300 days. The attained methane conversion efficiency based on the overall COD was 54% with an overall OLR of 0.39 g-COD·L^−1^·d^−1^ in the Na_2_SO_4_–NaHCO_3_ system, compared to 66% with at an OLR of 0.75 g-COD·L^−1^·d^−1^ in the NaCl system; the lower conversion efficiency was a result of SRB consumption. The acetate and propionate were detected in small amounts (less than 0.2 mg·L^−1^) after 55 days when the OLR was increased. The excess persisted for 60 days during which the concentration gradually decreased, possibly because of an increase in the microbial population. Comparing the relationship between the OLR and the system performance of the Na_2_SO_4_–NaHCO_3_ and NaCl systems, when the OLR increased to 0.6 g-COD·L^−1^·d^−1^, the propionate concentration in the NaCl system increased rapidly with a threshold shape (Figure 4), while in the Na_2_SO_4_–NaHCO_3_ system, the OLR increased to 0.8 g-COD·L^−1^·d^−1^, i.e., it did not increase significantly (Figure 5). The aforementioned phenomenon indicates that the propionate-utilizing microorganisms in the NaCl system were more sensitive than those in the Na_2_SO_4_–NaHCO_3_ system and there was a difference in the maximum uptake rate between the two systems. The hypothesis is confirmed from the kinetics values listed in Table 4.

### 3.2. Kinetic Parameter Estimation and Model Calibration

The experimentally obtained MPR, particulate COD, soluble COD, and VFA concentrations (acetate and propionate) were dynamically simulated as shown in Figure 4. The kinetic parameters were estimated by modifying the default values of the microbial reaction summarized in the anaerobic digestion model presented in Table 4 [18]. The calibrated specific disintegration rates (*k_dis_*) for the system were nearly two times the default values for typical solid waste because the experimental grass was pulverized and picked using a filter with a diameter of 220 μm, and the calibrated specific hydrolysis rates (*k_hyd,ch_, k_hyd,pr_* and *k_hyd,li_*) were comparable with those of grass-silage studies [35,36]. The value of the hydrolysis rate was less than default value, which can be attributed to different substrates resulting in accumulated biomass species discrepancies. Based on the obtained hydrolysis rate values, it is suggested that a smashing pretreatment is necessary to improve the synthetic halophyte anaerobic reaction rate. Some studies have also shown that using pretreatment technology can improve the biogas production of anaerobic fermentation [37,38]. The halophyte hydrolysis rate varies with the pulverization size and sodium concentration in the reactor. As shown in Table 4, the maximum uptake rates (*k_m,su_*, *k_m,aa_*, *k_m,fa_*, *k_m,va_*, *k_m,bu_*, *k_m,pro_*, *k_m,ac_* and *k_m,h2,_*) and the half-saturation coefficients (*K_S,su_*, *K_S,aa_*, *K_S,fa_*, *K_S,va_*, *K_S,bu_*, *K_S,pro_* and *K_S,ac_*) in the acidogenesis, acetogenesis, and methanogenesis processes were calibrated to reproduce the soluble concentration in the system, where the values were less than the default values owing to the high-density salt concentration [35,36,39,40], resulting in the estimated biomass species difference. In the acidogenesis process, the values of *k_m,su_*, and *k_m,aa_* were 30 d^−1^ and 50 d^−1^ in the ordinary anaerobic digestion process, while in the high salinity system, the values were 4 d^−1^.The maximum uptake rate of long-chain fatty acids (LCFA) was a limitation in the ordinary anaerobic process, where the value of *k_m,fa_* was 6 d^−1^. Under the high salinity in this study, the value of *k_m,fa_* was 1 d^−1^. Comparing the values of *k_m,su_,* and *k_m,aa_* (4 d^−1^), the LCFA uptake is still a limitation under high salinity. hence, in synthetic halophyte degradation, VFA accumulation should be considered. In the acetogenesis process, the *k_m,va_* and *k_m,bu_* values under high salinity (2 d^−1^) were 10 times smaller than those (20 d^−1^) in the ordinary condition. The *k_m,pro_* in the NaCl system was much smaller than the default value owing to the high sensitivity of the propionate degrader, and propionate had a sensitive inhibition in the propionate degrader with a *K_I,P,P_* of 800 gCOD·m^−3^ and a power factor (n) of 5. In this system, propionate inhibition also occurred in the acetate and hydrogen degraders with inhibition coefficients (*K_I,P,a_* and *K_I,P,h_*) of 500 gCOD·m^−3^. In the methanogenesis process, *k_m,ac_* and *k_m,h2_* in the high salinity system were less than those in the ordinary condition, which is similar to the situation in other processes. Furthermore, the half-saturation coefficients (*K_S_*) in this study were several times smaller than the default values owing to the biomass character depending on the species. From the parameter values obtained from the dynamic simulation in this study, the following conclusions could be obtained.

Methane production could be conducted under high salinity within a proper OLR;VFA accumulation (especially propionate) occurs over a certain OLR;The kinetics values under the salinity condition were smaller than the default values owing to the high sensitivity of the biomass characteristics, which are different from the ordinary ones.

The experimentally obtained datasets of the Na_2_SO_4_–NaHCO_3_ system were simulated as shown in Figure 5. The kinetics values are listed in Table 4. Except for several maximum uptake rates for propionate (*k_m,pro_*) and the sulfate-reduction parameters, the other kinetic parameters were the same as those in the NaCl system. The difference between the parameter values of the two systems was due to the substrates, which can result in different species in reactors, speculatively.

The model was used to simulate the process performance of a continuous anaerobic bioreactor with a mixed liquor suspended solids (MLSS) concentration of 10 g·L^−1^–40 g·L^−^^1^. In the NaCl system, the parameter values obtained from the dynamic simulation were used to conduct a steady-state simulation under continuous operation for 480 days. From the simulation, 66% of the fed COD was converted into gas in the reactor, as shown in Figure 6. The simulations indicate that the NaCl system could receive the influent up to a loading rate of 0.16 kg-COD·kg^−1^-MLSS·d^−1^ without significant deterioration of the methane conversion at 66%; this value was higher than the previously reported methane yield of 60% [41]. The converted fractions were observed to be methane gas. The particulate inert ratio was 27% from the lignin content in the influent. The biodegradable soluble component was 4%, which belonged to the digested organic after disintegration in the fermentation process; 3% of the input COD was transferred into biomass containing approximately 2% biodegradable particulate and 1% soluble inert.

### 3.3. Simulation Results and Model Verification

As shown in Figure 7, similar to the NaCl system, after steady-state simulation under continuous operation for 300 days, the Na_2_SO_4_–NaHCO_3_ system could receive more than 2 kg-COD·kg^−1^-MLSS·d^−1^, where 54% of the fed COD was converted into methane and another 12% was observed to be sulfide. The results showed that 66% of the influent COD was converted, of which 54% was methane gas and the remaining 12% was sulfide. The other component ratios were the same as those of the NaCl system.

The SRB parameter values were also obtained from the simulation by referring to the ones corresponding to the digestion-step kinetics values of the methane-fermentation process in the Na_2_SO_4_–NaHCO_3_ system. To highlight the need for the SRB particulate in the Na_2_SO_4_–NaHCO_3_ system model, the MPR simulation results with and without SRB were compared (see Figure 7). After 300 days of simulation, the MPR was 0.62 g-COD·L^−1^·d^−1^ without the SRB reaction at an OLR of 0.8 g-COD·L^−1^·d^−1^, while it was 0.35 g-COD·L^−1^·d^−1^ with the SRB particulate; thus, the simulation results were in agreement with the experimental datasets. According to the MPR simulation results with and without SRB, on day 300, approximately 43.5% of the degradable COD was consumed by the SRB.

## 4. Conclusions

We evaluated and modeled the anaerobic digestion of salt-accumulating plants including oxalate biodegradation and sulfate reduction. The following results were obtained.

The plant biomass under the 35.8 g-Na^+^·L^−1^ condition could be degraded in the anaerobic digestion reactor;The hydrolysis rate and maximum uptake rate in each step of NaCl and Na_2_SO_4_–NaHCO_3_ system anaerobic digestion were smaller than the default values owing to the species difference; the propionate uptake was a limited step for degradation in the NaCl system;A threshold propionate inhibition function with a power factor was developed on the propionate, acetate, and hydrogen degrader operating in the growth stage;In the NaCl system, 66% of the fed COD was degraded; this provides a biological post-treatment method for synthetic halophytes from phytoremediation;For the anaerobic digestion process in the Na_2_SO_4_–NaHCO_3_ system, 54% of the fed COD was converted into methane and another 12% was observed to be sulfide due to SRB.

## Figures and Tables

**Figure 1 ijerph-19-06943-f001:**
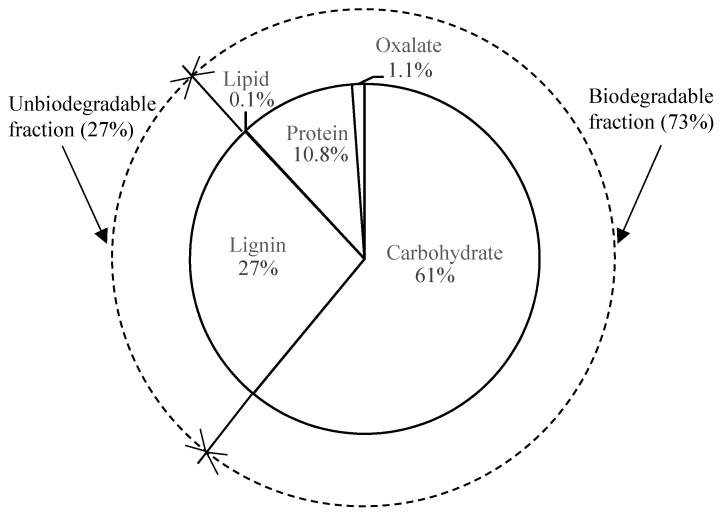
Dry fodder biomass component analysis results.

**Figure 2 ijerph-19-06943-f002:**
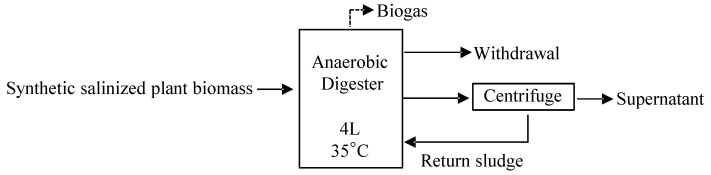
Continuous AD reactor operation flow.

**Figure 3 ijerph-19-06943-f003:**
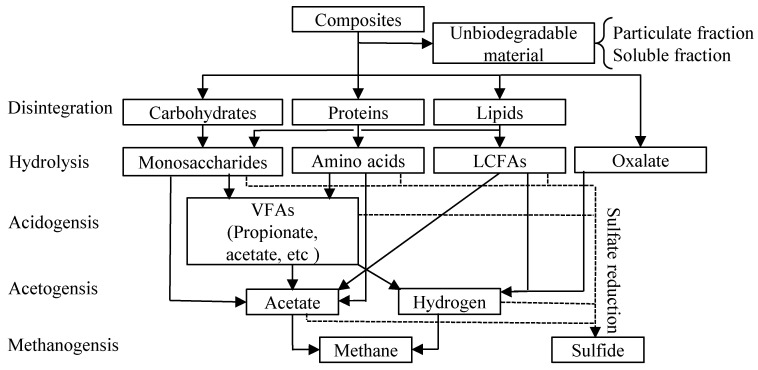
Organic decomposition of anaerobic fermentation including sulfate reduction, ―: methane production route, ---: sulfate-reduction route.

**Figure 4 ijerph-19-06943-f004:**
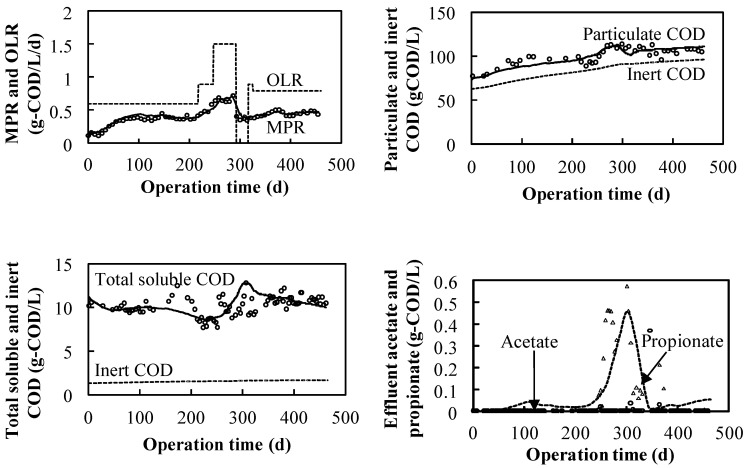
Measured and simulated results of MPR, VFAs, and particulate/soluble COD in the NaCl system.

**Figure 5 ijerph-19-06943-f005:**
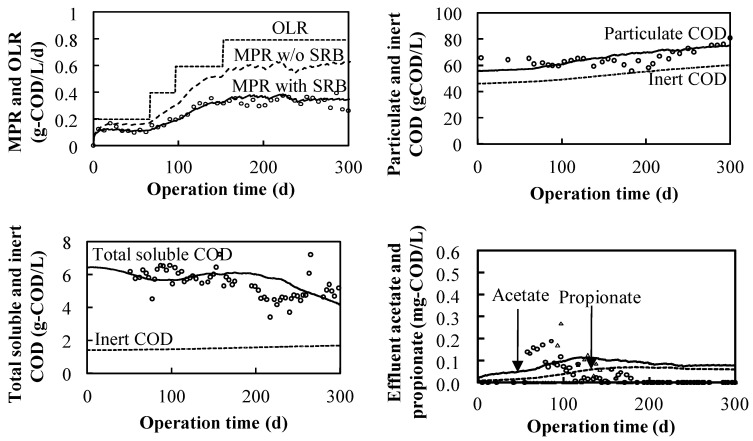
Measured and simulated results of MPR, VFAs, and particulate/soluble COD in the Na_2_SO_4_–NaHCO_3_ system.

**Figure 6 ijerph-19-06943-f006:**
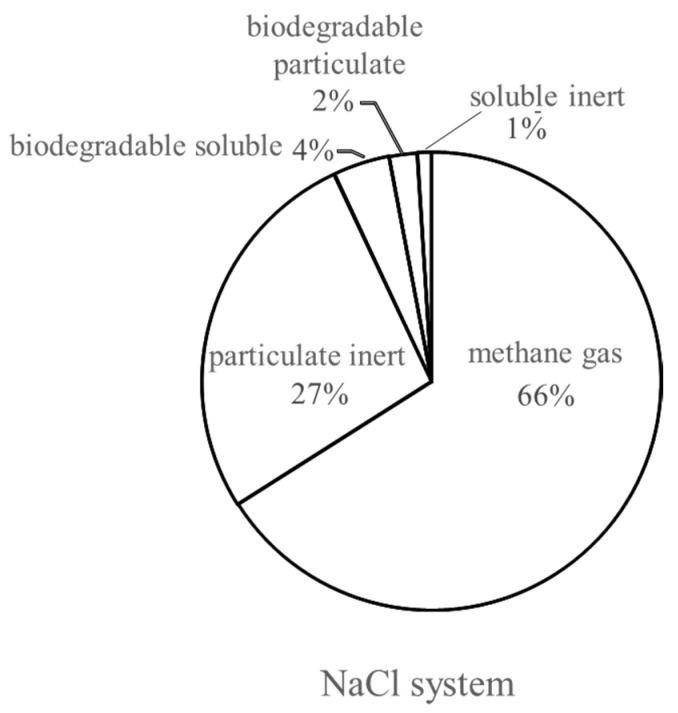
Conversion rate from the NaCl system in the steady state.

**Figure 7 ijerph-19-06943-f007:**
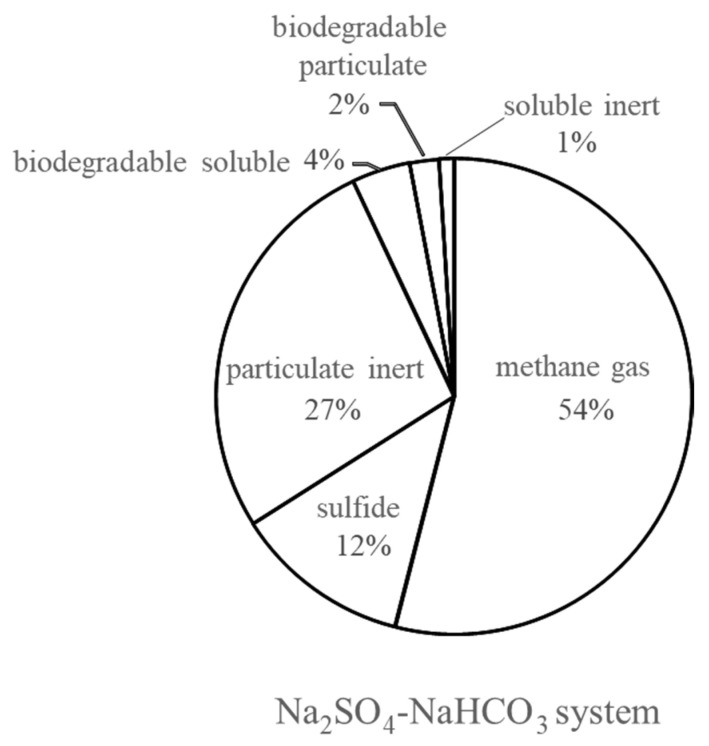
Conversion rate from the Na_2_SO_4_–NaHCO_3_ system in the steady state.

**Table 1 ijerph-19-06943-t001:** Feed substrates and conditions of the two systems.

	NaCl System	Na_2_SO_4_–Na_2_CO_3_ System	Na_2_SO_4_–Na_2_CO_3_ System (After Reaction)	Unit
Na^+^	35.83	27.6	27.6	g-Na·L^−1^
NaCl	70	-	56	mg·g^−1^
Na_2_SO_4_	-	34.84	-	mg·g^−1^
NaHCO_3_	-	31.52	-	mg·g^−1^
FeCl_2_⋅4H_2_O	-	54.18	5.96	mg·g^−1^
H_2_O	765.73	721.45	761.19	mg·g^−1^
Osmotic pressure	80.72	-	62.66	atm
Grass	158	158	158	g-COD·L^−1^

**Table 2 ijerph-19-06943-t002:** Gujer matrix for the anaerobic fermentation model of salt-accumulating plants including sulfate reduction for soluble components (i = 1–13; j = 1–37).

r	Component (i)→		1	2	3	4	5	6	7	8	9	10	11	12	13	Rate (ρ_j_) Type
	Process (j) ↓	Unit	*S_su_*	*S_aa_*	*S_fa_*	*S_va_*	*S_bu_*	*S_pro_*	*S_ac_*	*S_ox_*	*S_h2_*	*S_ch4_*	*S_h2s_*	*S_SO4_*	*S_I_*	
1	Disintegration	mgCOD·L^−1^·d^−1^													f_SI,xc_	F
2	Hydrolysis of carbohydrates	mgCOD·L^−1^·d^−1^	1													F
3	Hydrolysis of proteins	mgCOD·L^−1^·d^−1^		1												F
4	Hydrolysis of lipids	mgCOD·L^−1^·d^−1^	1 − *f_fa,li_*		*f_fa,li_*											F
5	Uptake of oxalate	mgCOD·L^−1^·d^−1^								−1	1					M
6	Uptake of monosaccharide	mgCOD·L^−1^·d^−1^	−1				(1 − *Y_su_*) × *f_bu,su1_*	(1 − *Y_su_*) × *f_pro,su1_*	(1 − *Y_su_*) × *f_ac,su1_*		(1 − *Y_su_*) × *f_h2,su1_*					M
7	Uptake of amino acids	mgCOD·L^−1^·d^−1^		−1		(1 − *Y_aa_*) × *f_va,aa1_*	(1 − *Y_aa_*) × *f_bu,aa1_*	(1 − *Y_aa_*) × *f_pro,aa1_*	(1 − *Y_aa_*) × *f_ac,aa1_*		(1 − *Y_aa_*) × *f_h2,aa1_*					M
8	Uptake of LCFA	mgCOD·L^−1^·d^−1^			−1				(1 − *Y_fa_*) × 0.7		(1 − *Y_fa_*) × 0.3					M
9	Uptake of valerate	mgCOD·L^−1^·d^−1^				−1		(1 − *Y_c4_*) × 0.54	(1 − *Y_c4_*) × 0.31		(1 − *Y_c4_*) × 0.15					M
10	Uptake of butyrate	mgCOD·L^−1^·d^−1^					−1		(1 − *Y_c4_*) × 0.8		(1 − *Y_c4_*) × 0.2					M
11	Uptake of propionate	mgCOD·L^−1^·d^−1^						−1	(1 − *Y_pro_*) × 0.57		(1 − *Y_pro_*) × 0.43					M∙I
12	Uptake of acetate	mgCOD·L^−1^·d^−1^							−1			1 − *Y_ac_*				M∙I
13	Uptake of hydrogen	mgCOD·L^−1^·d^−1^									−1	1 − *Y_h2_*				M∙I
14	Uptake of monosaccharide by SRB	mgCOD·L^−1^·d^−1^	−1				(1 − *Y_mSBR_*) × *f_bu,su2_*	(1 − *Y_mSBR_*) × *f_pro,su2_*	(1 − *Y_mSBR_*) × *f_ac,su2_*				(1 − *Y_mSRB_*) × *f_h2s,su_*/64	−(1 − *Y_mSRB_*) × *f_h2s,su_*/64		M
15	Uptake of amino acid by SRB	mgCOD·L^−1^·d^−1^		−1		(1 − *Y_aaSRB_*) × *f_va,aa2_*	(1 − *Y_aaSRB_*) × *f_bu,aa2_*	(1 − *Y_aaSRB_*) × *f_pro,aa2_*	(1 − *Y_aaSRB_*) × *f_ac,aa2_*				(1 − *Y_aaSRB_*) × *f_h2s,aa_*/64	−(1 − *Y_aaSRB_*) × *f_h2s,aa_*/64		M
16	Uptake of LCFA by SRB	mgCOD·L^−1^·d^−1^			−1				(1 − *Y_LSRB_*) × *f_ac,L_*				(1 − *Y_LSRB_*) × (1 − *f_ac,L_*)/64	−(1 − *Y_LSRB_*) × (1 − *f_ac,L_*)/64		M
17	Uptake of valerate by SRB	mgCOD·L^−1^·d^−1^				−1			(1 − *Y_vSRB_*) × 0.84				(1 − *Y_vSRB_*) × 0.16/64	−(1 − *Y_vSRB_*) × 0.16/64		M
18	Uptake of butyrate by SRB	mgCOD·L^−1^·d^−1^					−1		(1 − *Y_bSRB_*) × 0.8				(1 − *Y_bSRB_*) × 0.2/64	−(1 − *Y_bSRB_*) × 0.2/64		M
19	Uptake of propionate by SRB	mgCOD·L^−1^·d^−1^						−1	(1 − *Y_pSRB_*) × 0.57				(1 − *Y_pSRB_*) × 0.43/64	−(1 − *Y_pSRB_*) × 0.43/64		M
20	Uptake of acetate by SRB	mgCOD·L^−1^·d^−1^							−1				(1 − *Y_aSRB_*)/64	−(1 − *Y_aSRB_*)/64		M
21	Uptake of hydrogen by SRB	mgCOD·L^−1^·d^−1^									−1		(1 − *Y_hSRB_*)/64	−(1 − *Y_hSRB_*)/64		M
22	Decay of X_ox_	mgCOD·L^−1^·d^−1^														F
23	Decay of X_su_	mgCOD·L^−1^·d^−1^														F
24	Decay of X_aa_	mgCOD·L^−1^·d^−1^														F
25	Decay of X_fa_	mgCOD·L^−1^·d^−1^														F
26	Decay of X_c4_	mgCOD·L^−1^·d^−1^														F
27	Decay of X_pro_	mgCOD·L^−1^·d^−1^														F
28	Decay of X_ac_	mgCOD·L^−1^·d^−1^														F
29	Decay of X_h2_	mgCOD·L^−1^·d^−1^														F
30	Decay of X_mSRB_	mgCOD·L^−1^·d^−1^														F
31	Decay of X_aaSRB_	mgCOD·L^−1^·d^−1^														F
32	Decay of X_LSRB_	mgCOD·L^−1^·d^−1^														F
33	Decay of X_vSRB_	mgCOD·L^−1^·d^−1^														F
34	Decay of X_bSRB_	mgCOD·L^−1^·d^−1^														F
35	Decay of X_pSRB_	mgCOD·L^−1^·d^−1^														F
36	Decay of X_aSRB_	mgCOD·L^−1^·d^−1^														F
37	Decay of X_hSRB_	mgCOD·L^−1^·d^−1^														F
	Biomass Yield (gCOD·gCOD^−1^) Ysu = 0.18 Yaa = 0.18 Yfa = 0.06 Yc4 = 0.04 Ypro = 0.04 Yac = 0.05 Yh2 = 0.04		Monosaccharides (kgCOD·m^−3^)	Amino acids (kgCOD·m^−3^)	Long-chain fatty acids (kgCOD·m^−3^)	Total valerate (kgCOD·m^−3^)	Total butyrate (kgCOD·m^−3^)	Total propionate (kgCOD·m^−3^)	Total acetate (kgCOD·m^−3^)	Oxalate (kgCOD·m^−3^)	Hydrogen gas (kgCOD·m^−3^)	Methane gas (kgCOD·m^−3^)	Hydrogen sulfide (kmol·m^−3^)	Total sulfates (kmol·m^−3^)	Soluble inerts (kgCOD·m^−3^)	

**Table 3 ijerph-19-06943-t003:** Gujer matrix for the anaerobic fermentation model of salt-accumulating plants including sulfate reduction for particulate components (i = 14–34; j = 1–37).

r	Component (i)→		14	15	16	17	18	19	20	21	22	23	24	25	26	27	28	29	30	31	32	33	34	Rate (ρ_j_) Type
	Process (j)↓	Unit	*X_C_*	*X_ch_*	*X_pr_*	*X_li_*	*X_I_*	*X_ox_*	*X_su_*	*X_aa_*	*X_fa_*	*X_c4_*	*X_pro_*	*X_ac_*	*X_h2_*	*X_mSRB_*	*X_aaSRB_*	*X_LSRB_*	*X_vSRB_*	*X_bSRB_*	*X_pSRB_*	*X_aSRB_*	*X_hSRB_*	
1	Disintegration	mgCOD·L^−1^·d^−1^	−1	*f_ch,xc_*	*f_pr,xc_*	*f_li,xc_*	*f_XI,xc_*																	F
2	Hydrolysis of carbohydrates	mgCOD·L^−1^·d^−1^		−1																				F
3	Hydrolysis of proteins	mgCOD·L^−1^·d^−1^			−1																			F
4	Hydrolysis of lipids	mgCOD·L^−1^·d^−1^				−1																		F
5	Uptake of oxalate	mgCOD·L^−1^·d^−1^						*Y_ox_*																M
6	Uptake of monosaccharide	mgCOD·L^−1^·d^−1^							*Y_su_*															M
7	Uptake of amino acids	mgCOD·L^−1^·d^−1^								*Y_aa_*														M
8	Uptake of LCFA	mgCOD·L^−1^·d^−1^									*Y_fa_*													M
9	Uptake of valerate	mgCOD·L^−1^·d^−1^										*Y_c4_*												M
10	Uptake of butyrate	mgCOD·L^−1^·d^−1^										*Y_c4_*												M
11	Uptake of propionate	mgCOD·L^−1^·d^−1^											*Y_pro_*											M
12	Uptake of acetate	mgCOD·L^−1^·d^−1^												*Y_ac_*										M∙I
13	Uptake of hydrogen	mgCOD·L^−1^·d^−1^													*Y_h2_*									M∙I
14	Uptake of monosaccharide by SRB	mgCOD·L^−1^·d^−1^														*Y_mSRB_*								M∙I
15	Uptake of amino acid by SRB	mgCOD·L^−1^·d^−1^															*Y_aaSRB_*							M
16	Uptake of LCFA by SRB	mgCOD·L^−1^·d^−1^																*Y_LSRB_*						M
17	Uptake of valerate by SRB	mgCOD·L^−1^·d^−1^																	*Y_vSRB_*					M
18	Uptake of butyrate by SRB	mgCOD·L^−1^·d^−1^																		*Y_bSRB_*				M
19	Uptake of propionate by SRB	mgCOD·L^−1^·d^−1^																			*Y_pSRB_*			M
20	Uptake of acetate by SRB	mgCOD·L^−1^·d^−1^																				*Y_aSRB_*		M
21	Uptake of hydrogen by SRB	mgCOD·L^−1^·d^−1^																					*Y_hSRB_*	M
22	Decay of X_ox_	mgCOD·L^−1^·d^−1^						−1																F
23	Decay of X_su_	mgCOD·L^−1^·d^−1^	1						−1															F
24	Decay of X_aa_	mgCOD·L^−1^·d^−1^	1							−1														F
25	Decay of X_fa_	mgCOD·L^−1^·d^−1^	1								−1													F
26	Decay of X_c4_	mgCOD·L^−1^·d^−1^	1									−1												F
27	Decay of X_pro_	mgCOD·L^−1^·d^−1^	1										−1											F
28	Decay of X_ac_	mgCOD·L^−1^·d^−1^	1											−1										F
29	Decay of X_h2_	mgCOD·L^−1^·d^−1^	1												−1									F
30	Decay of X_mSRB_	mgCOD·L^−1^·d^−1^	1													−1								F
31	Decay of X_aaSRB_	mgCOD·L^−1^·d^−1^	1														−1							F
32	Decay of X_LSRB_	mgCOD·L^−1^·d^−1^	1															−1						F
33	Decay of X_vSRB_	mgCOD·L^−1^·d^−1^	1																−1					F
34	Decay of X_bSRB_	mgCOD·L^−1^·d^−1^	1																	−1				F
35	Decay of X_pSRB_	mgCOD·L^−1^·d^−1^	1																		−1			F
36	Decay of X_aSRB_	mgCOD·L^−1^·d^−1^	1																			−1		F
37	Decay of X_hSRB_	mgCOD·L^−1^·d^−1^	1																				−1	F
	fch,xc = 0.61 fh2,su = 0.33 fpr,xc = 0.11 fva,aa = 0.26 fli,xc = 0.01 fbu,aa = 0.27 fXI,xc = 0.27 fpro,aa = 0.07 fL,xc = 0.001 fac,aa = 0.33 fbu,su = 0 fh2,aa = 0.07 fpro,su =0 fac,L = 0.7 fpro,ac = 0.67 f values are same in SRB		Composites (kgCOD·m^−3^)	Carbohydrates (kgCOD·m^−3^)	Proteins (kgCOD·m^−3^)	Lipids (kgCOD·m^−3^)	Inert (kgCOD·m^−3^)	Oxalate degraders (kgCOD·m^−3^)	Sugar degraders (kgCOD·m^−3^)	Amino acid (kgCOD·m^−3^)	LCFA degraders (kgCOD·m^−3^)	Valerate and butyrate degraders (kgCOD·m^−3^)	Propionate degraders (kgCOD·m^−3^)	Acetate degraders (kgCOD·m^−3^)	Hydrogen degraders (kgCOD·m^−3^)	SRB from monosaccharide (kgCOD·m^−3^)	SRB from amino acid (kgCOD·m^−3^)	SRB from LCFA (kgCOD·m^−3^)	SRB from valerate (kgCOD·m^−3^)	SRB from butyric (kgCOD·m^−3^)	SRB from propionate (kgCOD·m^−3^)	SRB from acetate (kgCOD·m^−3^)	SRB from hydrogen (kgCOD·m^−3^)	

**Table 4 ijerph-19-06943-t004:** Kinetics for the anaerobic fermentation model of salt-accumulating plants including sulfate reduction.

Item	Symbol	Default Value	NaCl System	Na_2_SO_4_–NaHCO_3_ System	Unit
Disintegration					
Disintegration rate	*k_dis_*	0.5	1.2	1.2	d^−1^
Hydrolysis					
Carbohydrate hydrolysis rate	*k_hyd,ch_*	10	10	10	d^−1^
Protein hydrolysis rate	*k_hyd,pr_*	10	10	10	d^−1^
Lipids hydrolysis rate	*k_hyd,li_*	10	10	10	d^−1^
Acidogenesis					
Maximum uptake rate by oxalate degrader	*k_m,ox_*				
Half saturation coefficient of oxalate degrader	*K_S,ox_*				
Specific decay rate of oxalate degrader	*b_ox_*				
Maximum uptake rate by sugar degrader	*k_m,su_*	30	4	4	d^−1^
Half saturation coefficient of sugars degrader	*K_S,su_*	500	10	10	gCOD·m^−3^
Specific decay rate of sugars degrader	*b_su_*	-	0.06	0.06	d^−1^
Maximum uptake rate by amino-acids degrader	*k_m,aa_*	50	4	4	d^−1^
Half saturation coefficient of amino-acids degrader	*K_S,aa_*	300	10	10	gCOD·m^−3^
Specific decay rate of amino-acids degrader	*b_aa_*	-	0.06	0.06	d^−1^
Maximum uptake rate by LCFAs degrader	*k_m,fa_*	6	1	1	d^−1^
Half-saturation coefficient of LCFAs degrader	*K_S,fa_*	400	40	40	gCOD·m^−3^
Specific decay rate of LCFAs degrader	*b_fa_*	-	0.06	0.06	d^−1^
Acetogenesis					
Maximum uptake rate by valerate degrader	*k_m,va_*	20	2	2	d^−1^
Half-saturation coefficient of valerate degrader	*K_S,va_*	200	10	10	gCOD·m^−3^
Specific decay rate of valerate and butyrate degrader	*b_c4_*	-	0.06	0.06	d^−1^
Maximum uptake rate by butyrate degrader	*k_m,bu_*	20	2	2	d^−1^
Half-saturation coefficient of butyrate degrader	*K_S,bu_*	200	10	10	gCOD·m^−3^
Maximum uptake rate by propionate degrader	*k_m,pro_*	13	0.039	2	d^−1^
Half-saturation coefficient of propionate degrader	*K_S,pro_*	100	5	5	gCOD·m^−3^
Propionate inhibition coefficient on propionate degrader	*K_I,p,p_*	-	800	800	gCOD·m^−3^
Specific decay rate of propionate degrader	*b_pro_*	-	0.06	0.06	d^−1^
Propionate inhibition power coefficient	*n*	-	5	5	-
Methanogenesis					
Maximum uptake rate by acetate degrader	*k_m,ac_*	8	4	4	d^−1^
Half-saturation coefficient of acetate degrader	*K_S,ac_*	150	15	15	gCOD·m^−3^
Propionate inhibition coefficient on acetate degrader	*K_I,p,a_*	-	500	500	gCOD·m^−3^
Maximum uptake rate by hydrogen degrader	*k_m,h2_*	35	1.5	1.5	d^−1^
Half saturation coefficient of hydrogen degrader	*K_S,h2_*	7 × 10^−6^	7 × 10^−6^	7 × 10^−6^	gCOD·m^−3^
Propionate inhibition coefficient on hydrogen degrader	*K_I,p,h_*	-	500	500	gCOD·m^−3^
Sulfate reduction					
SRB maximum specific growth rate of sugar degrader	*k_m,SRB_*	-	-	2	d^−1^
SRB half-saturation coefficient of sugars degrader	*K_S,m,SRB_*	-		0.1	gCOD·m^−3^
SRB specific decay rate of sugars degrader	*b_mSRB_*	-	-	0.06	d^−1^
SRB maximum specific growth rate of amino acids degrader	*k_aa,SRB_*		-	2	d^−1^
SRB half-saturation coefficient of amino acids degrader	*K_S,aa,SRB_*	-	-	0.1	gCOD·m^−3^
SRB specific decay rate of amino acids degrader	*b_aaSRB_*	-	-	0.06	d^−1^
SRB maximum specific growth rate of LCFAs degrader	*k_L,SRB_*	-	-	1	d^−1^
SRB half-saturation coefficient of LCFAs degrader	*K_S,L,SRB_*	-	-	0.1	gCOD·m^−3^
SRB specific decay rate of LCFAs degrader	*b_faSRB_*	-	-	0.06	d^−1^
SRB maximum specific growth rate of valerate degrader	*k_v,bu,SRB_*	-	-	2	d^−1^
SRB half-saturation coefficient of valerate degrader	*K_S,v,SRB_*	-	-	0.1	gCOD·m^−3^
SRB specific decay rate of valerate degrader	*b_vSRB_*	-	-	0.06	d^−1^
SRB maximum specific growth rate of butyrate degrader	*k_m,bu,SRB_*	-	-	2	d^−1^
SRB half-saturation coefficient of butyrate degrader	*K_S,bu,SRB_*	-	-	0.1	gCOD·m^−3^
SRB specific decay rate of butyrate degrader	*b_vSRB_*	-	-	0.06	d^−1^
SRB maximum specific growth rate of propionate degrader	*k_m,pro,SRB_*	-	-	2	d^−1^
SRB half-saturation coefficient of propionate degrader	*K_S,pro,SRB_*	-	-	0.1	gCOD·m^−3^
SRB specific decay rate of propionate degrader	*b_pSRB_*	-	-	0.06	d^−1^
SRB maximum specific growth rate of acetate degrader	*k_m,ac,SRB_*	-	-	2	d^−1^
SRB half-saturation coefficient of acetate degrader	*K_S,ac,SRB_*	-	-	0.1	gCOD·m^−3^
SRB specific decay rate of acetate degrader	*b_aSRB_*	-	-	0.06	d^−1^
SRB maximum specific growth rate of hydrogen degrader	*k_m,h2,SRB_*	-	-	8	d^−1^
SRB half-saturation coefficient of hydrogen degrader	*K_S,h2,SRB_*	-	-	0.1	gCOD·m^−3^
SRB specific decay rate of hydrogen degrader	*b_hSRB_*	-	-	0.06	d^−1^

## Data Availability

Not applicable.

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
