# Peer review of "A Kinetic Model for Anaerobic Digestion and Biogas Production of Plant Biomass under High Salinity"

_ijerph, 2022, doi:10.3390/ijerph19116943_

Round 1
Reviewer 1 Report
Manuscript
Title: „A kinetic model for anaerobic digestion and biogas production of plant biomass under high salinity”
Authors: Jing Wang, Bing Liu, Meng Sun, Feiyong Chen, Mitsuharu Terashima, Hidenari Yasui
Dear Authors
I revised the manuscript: " A kinetic model for anaerobic digestion and biogas production of plant biomass under high salinity submitted to the “Internationa Journal of Environmental Research and Public Health” Journal. The paper is very interesting. However, I have some concerns, which need to be addressed.
Line 2-3. The theme of the article is logical and reflects well the issues addressed.
Abstract:
Line 13-27.The abstract contains the necessary information and correctly presents the scientific achievements of the authors.
Line 16, 18. „….at 35.8 g-Na+ /L…..” Indication of the unit of measure " g-Na+ /L”, using a fractional dash is acceptable but is colloquial in meaning. However, exponential notation should be used.
Please use the notation of the quotient in units of measure using mathematical notation with a power exponent for example: g-Na+·L-1
Line 17, 18. „….0.58–1.5 g-COD/L/d….” Please use the notation of the quotient in units of measure using mathematical notation with a power exponent for example: g-COD·L-1 ·d-1 .
The unit of measure should be present with both range values.
Line 19, 26. „….COD…..” Please explain the introduced abbreviations of terms, possibly as soon as they are introduced in the content of the article. A single explanation is sufficient in the execution of the explanation.
Line 23. „….of 10–40 g/L…..” Please use the notation of the quotient in units of measure using mathematical notation with a power exponent for example: g·L-1.
The unit of measure should be present with both range values.
Line 23. „….MLSS….” Please explain the introduced abbreviations of terms, possibly as soon as they are introduced in the content of the article. A single explanation is sufficient in the execution of the explanation.
Line 24, 25. „….loading rate of 0.16 kg-COD/kg-MLSS/d….”….” Please use the notation of the quotient in units of measure using mathematical notation with a power exponent for example: kg-COD·kg-MLSS-1 ·d-1 .
Keywords: There is a visible lack of the keyword " of high substrate salinity" because this is the word exposed in the topic of the article. Please consider completing the statement.
1. Introduction
The content of the chapter presents well the state of scientific knowledge and the genesis of the research topic. The introduction of information about the goal and scope of the work is laconic and should be expanded adequately to the adopted course of research.
Line 50. „….(70 g-NaCl/L)….” Please use the notation of the quotient in units of measure using mathematical notation with a power exponent for example: g-NaCl·L-1
Line 63-64. Lack of table name and lack of assigned table numer.
2. Materials and methods
The presented methodological information is extremely precise. The methodology used and the research stands are well described. The division into subsections is logical and represents well the complexity of the research. There is a lack of information on reference samples to indicate the effects of experiments. Identifying the role of the reference sample is crucial in the experiment. Please highlight this aspect in the chapter.
Line 80. Figure 1. The figure is readable and supports the information in the text.
Line 85,86. Please use the notation of the quotient in units of measure using mathematical notation with a power exponent for example: g-COD·L-1 , g-grass·L-1 .
Line 88, 89. „…was 70 mg-NaCl/g….” Please use the notation of the quotient in units of measure using mathematical notation with a power exponent for example: mg-NaCl·g-1 .
Line 93. „….at 35.8 g-Na+/L and 27.6 g-Na+/L…..” Please use the notation of the quotient in units of measure using mathematical notation with a power exponent for example: g-Na+·L-1
Line 94. „….35oC….” The unit of measure and value should be separated by a space: 35 oC.
Line 96. „….(1 mg-Ni/L and 1 mg-Co/L)…..” Please use the notation of the quotient in units of measure using mathematical notation with a power exponent for example: mg-Ni·L-1 , mg-Co·L-1 .
Line 98 Table 1. Please update the numbering of the tables.
„…..g-Na/L….., ……mg/g….., …..g-COD/L….” Please use the notation of the quotient in units of measure using mathematical notation with a power exponent for example: g-Na·L-1 , mg·g-1 , g-COD·L-1 .
Line 101, 103, 104. „….0.58 g-COD/L/d….” Please use the notation of the quotient in units of measure using mathematical notation with a power exponent for example: g-COD·L-1 ·d-1 .
Line 103, 104. „….0.58–1.50 g-COD/L/d…..” Please use the notation of the quotient in units of measure using mathematical notation with a power exponent for example: g-COD·L-1 ·d-1 . The unit of measure should be present with both range values.
Line 112 Figure 2 "Artificial plant biomass" is an extremely misleading phrase. I suggest replacing this mental shortcut with a more fitting statement.
Line 117 „…to 1 mL/min….” Please use the notation of the quotient in units of measure using mathematical notation with a power exponent for example: mL·min-1.
Line 118„….with 4 mol/L…..” Please use the notation of the quotient in units of measure using mathematical notation with a power exponent for example: mol·L-1.
Line 137. „….using 1.19 g-COD particulate/g-VSS…..” Please use the notation of the quotient in units of measure using mathematical notation with a power exponent for example: g-COD particulate·g-VSS-1.
Line 138. „….and 2.67 g-COD soluble/g-TOC…..” Please use the notation of the quotient in units of measure using mathematical notation with a power exponent for example: g-COD soluble·g-TOC-1.
Line 138, 139. „….32 g-O2/12 g-C)…..” Please use the notation of the quotient in units of measure using mathematical notation with a power exponent for example: 32 g-O2·12 g-C-1.
Line 149. ".....COD/DOC factors ......" If the notation is not related to units of measurement then you can leave the ratio notation with a slash or use a ":" colon.
Line 149. „…..(g-COD/g-DOC)…..” Please use the notation of the quotient in units of measure using mathematical notation with a power exponent for example: g-COD·g-DOC-1.
Line 151. "....as 2.67, 3.00, and 2.92, ....." Are the units of measurement known?
Line 153. Figure 3. The figure is clear but has an incomprehensible legend placed in the figure name.
Line 164. „…..SRB……” Please explain the introduced abbreviations of terms, possibly as soon as they are introduced in the content of the article. A single explanation is sufficient in the execution of the explanation.
Line 167, 168. „….(ρj = kprocess∙XB and ρj = bbiomass∙XB) (in r1-r4 and r22-r37) and Monod-type (ρj = km∙S/(KS+S)∙XB) (in r5-r11 and r15-r21) rate equations…..”Symbols require textual description and explanations. Mathematically correct notation of formulas and correlations should be generated in special sub-programmes. Mathematical equations with significance for research results should be assigned sequential numbers.
Line 172. „….(ρj = Ktn /(Ktn +Stin ))….” Symbols require textual description and explanations. Mathematically correct notation of formulas and correlations should be generated in special sub-programmes. Mathematical equations with significance for research results should be assigned sequential numbers.
Line 172. „…..r12, r13, r14……” Symbols require textual description and explanations.
Line 175-177. Table 2….. mgCOD/L/d…. Please use the notation of the quotient in units of measure using mathematical notation with a power exponent for example: mg-COD·L-1·d-1
Line 175. „….LCFA….” Please explain the introduced abbreviations of terms, possibly as soon as they are introduced in the content of the article. A single explanation is sufficient in the execution of the explanation.
Line 175, Table 2. row „Component” …. Symbolic designations require a text description
Line 175, Table 2, "....Rate, Type....M...F..." symbols require textual description and explanations
Line 175-177, Tabel 2.:
„….kgCOD/m3 …….” Please use the notation of the quotient in units of measure using mathematical notation with a power exponent for example: kg-COD·m-3 .
„….gCOD/gCOD…..” Please use the notation of the quotient in units of measure using mathematical notation with a power exponent for example: g-COD·g-COD-1.
Line 178, 180. Table 3.:
„….mgCOD/L/d…..” Please use the notation of the quotient in units of measure using mathematical notation with a power exponent for example: mg-COD·L-1·d-1 .
„…..kgCOD/m3…..” Please use the notation of the quotient in units of measure using mathematical notation with a power exponent for example: kg-COD·m-3 .
"....Rate (ρj) Type..... F, M, M-I.... and so on - Please indicate the explanations for the abbreviations and for the letter designations in this column. This will enable better interpretation of the information by the reader.
Line 185. „….27.6 or 35.8g-Na+/L…” Lack of space between value and unit of measurement. Please correct this.Please use the notation of the quotient in units of measure using mathematical notation with a power exponent for example: g-Na+·L-1.
The unit of measure should be present with both range values: 27.6 g-Na+·L-1 or 35.8 g-Na+·L-1.
3. Results and discussion
The discussion of the results is conducted in a typical manner for the undertaken research topic. The results are adequate to the research assumptions presented.
Line 191. „…..from 0.58 to 1.50 g-COD/L/d …..”….” Please use the notation of the quotient in units of measure using mathematical notation with a power exponent for example: g-COD·L-1 .
The unit of measure should be present with both range values: 0.58 g-COD·L-1 to 1.50 g-COD·L-1.
Line 198. „….at 21.5 g/L…..” Please use the notation of the quotient in units of measure using mathematical notation with a power exponent for example: g·L-1.
Line 199. „….70 g/L….” Please use the notation of the quotient in units of measure using mathematical notation with a power exponent for example: g·L-1.
Line 203. „….to 0.8 g-COD/L/d and changed to 0.75 g-COD/L/d….” Please use the notation of the quotient in units of measure using mathematical notation with a power exponent for example: g-COD·L-1 ·d-1 .
Line 204. „…0.75 g-COD/L/d. ….” Please use the notation of the quotient in units of measure using mathematical notation with a power exponent for example: g-COD·L-1 ·d-1 .
Line 206. Figure 4.:
„….g-COD/L….”….” Please use the notation of the quotient in units of measure using mathematical notation with a power exponent for example: g-COD·L-1 .
„…..g-COD/L/d…..”….” Please use the notation of the quotient in units of measure using mathematical notation with a power exponent for example: g-COD·L-1 ·d-1 .
Line 210. „….between 0.2 and 0.8 g-COD/L/d….” Please use the notation of the quotient in units of measure using mathematical notation with a power exponent for example: g-COD·L-1 ·d-1 . The unit of measure should be present with both range values
Line 211, 212. „….54 %....” The form of the value and the percentage symbol are always written without spaces. Please correct this.
Line 212, 218, 220. „….of 0.39 g-COD/L/d….” Please use the notation of the quotient in units of measure using mathematical notation with a power exponent for example: g-COD·L-1 ·d-1 .
Line 214-215. „…..less than 0.2 mg/L…..” Please use the notation of the quotient in units of measure using mathematical notation with a power exponent for example: mg·L-1.
Line 226. Table 4. „….gCOD/m3…..” Please use the notation of the quotient in units of measure using mathematical notation with a power exponent for example: g-COD·m-3.
Line 226. Table 4. „….d-1….” Please maintain a consistent form of notation for the unit of measurement: „d-1”.
Line 226. „…SRB….” Please explain the introduced abbreviations of terms, possibly as soon as they are introduced in the content of the article. A single explanation is sufficient in the execution of the explanation.
Line 167, 168, 172, Table 2 175, Table 3 178, Table 4 226, 244,245, and similar „….(km,su, km,aa, km,fa, km,va, km,bu, km,pro, km,ac and km,h2,) and the half saturation coefficients (KS,su, KS,aa, KS,fa, KS,va, KS,bu, KS,pro and KS,ac)…..” The signs and symbols used in the paper should be clearly explained when they appear in the stabelled information and in the text. The huge portion of this message overwhelms the reader despite the fact that the authors of the article think that this system of signs is obvious.
Line 260, 262 „….of 800 gCOD/m3…..” Please use the notation of the quotient in units of measure using mathematical notation with a power exponent for example: g-COD·m-3 .
Line 281. Figure 6.:
„….g-COD/L….” Please use the notation of the quotient in units of measure using mathematical notation with a power exponent for example: g-COD·L-1 .
„…..g-COD/L/d…..” Please use the notation of the quotient in units of measure using mathematical notation with a power exponent for example: g-COD·L-1 ·d-1 .
Line 286. „….of 10–40 g/L….” Please use the notation of the quotient in units of measure using mathematical notation with a power exponent for example: g·L-1.
The unit of measure should be present with both range values: 10 g·L-1 -40 g·L-1.
Line 290, 302. „….rate of 0.16 kg-COD/kg-MLSS/d….”….”….” Please use the notation of the quotient in units of measure using mathematical notation with a power exponent for example: kg-COD·kg-MLSS-1 ·d-1 .
Line 310, 311. „….was 0.62 g-COD/L/d….” Please use the notation of the quotient in units of measure using mathematical notation with a power exponent for example: g-COD·L-1 ·d-1 .
4. Conclusions
Line 319. „….under the 35.8 g-Na+/L….” Please use the notation of the quotient in units of measure using mathematical notation with a power exponent for example: g-Na+·L-1 .
In the chapter a description of the obtained results dominates. Please consider introducing additionally information which really constitute conclusions on the basis of the obtained results. A good solution is to formulate a research hypothesis and validate it.
Reviewer 2 Report
- The keywords should be more specific
- The novelty of the study is not clear; Please give some insights regarding this in the aims paragraph
- Line 84: “completely mixed lab-scale (…) reactors” – what are the mixing conditions? Mechanical agitation? Agitation rate?
- Table 1: there is no need for a separate column for units; these units can be added to the parameter
- Line 114: what is (are) the detector(s) used for cation and anion determination?
- Line 120: why the sugars concentration was not quantified by HPLC?
- Line 128: In my opinion, the methodology for lignin quantification should be more detailed
- Why Figure 5 appears before Figures 6 and 7 if in the text it is mentioned only in section 3.3?
- Line 316: it seems that the 2nd sentence is not finished
